# Design of high-affinity binders to immune modulating receptors for cancer immunotherapy

Wei Yang[1,2], Derrick R. Hicks[1,2], Agnidipta Ghosh [3], Tristin A. Schwartze[4], Brian Conventry [1,2], Inna Goreshnik[1,2], Aza Allen[1,2], Samer F. Halabiya[1,2], Chan Johng Kim[1,2], Cynthia S. Hinck[4], David S. Lee [1,2], Asim K. Bera [1,2], Zhe Li[1,2], Yujia Wang[1,2], Thomas Schlichthaerle[1,2], Longxing Cao[1,2], Buwei Huang[1,2], Sarah Garrett[3], Stacey R. Gerben [1,2], Stephen Rettie[1,2], Piper Heine[1,2], Analisa Murray[1,2], Natasha Edman [1,2], Lauren Carter[1,2], Lance Stewart [1,2], Steven C. Almo[3], Andrew P. Hinck [4] & David Baker [1,2,5] ✉

Immune receptors have emerged as critical therapeutic targets for cancer immunotherapy. Designed protein binders can have high affinity, modularity, and stability and hence could be attractive components of protein therapeutics directed against these receptors, but traditional Rosetta based protein binder methods using small globular scaffolds have difficulty achieving high affinity on convex targets. Here we describe the development of helical concave scaffolds tailored to the convex target sites typically involved in immune receptor interactions. We employed these scaffolds to design proteins that bind to TGFβRII, CTLA-4, and PD-L1, achieving low nanomolar to picomolar affinities and potent biological activity following experimental optimization. Co-crystal structures of the TGFβRII and CTLA-4 binders in complex with their respective receptors closely match the design models. These designs should have considerable utility for downstream therapeutic applications.

Receptors expressed on immune and tumor cells are crucial in maintaining homeostasis by up regulating or down regulating the immune response[1]. Cancer immunotherapy has revolutionized the field of oncology by prolonging the survival of patients with cancer over the past decades[2]. Inhibitory receptors such as CTLA-4, PD-1, LAG3, and PD-L1, which regulate immune responses, alongside cytokine receptors like il2-R, il10-R, and TGFbRII, which control the proliferation and differentiation of immune cells, have become important therapeutic targets for cancer immunotherapy[3,4]. The development of computational protein design has enabled the creation of de novo protein binders against human cell surface receptors[5–8] with high modularity, affinity, and stability[9–12]. Given the complexity of the immunomodulatory network, and the growing interest in targeting multiple receptors in combination therapies, high-affinity and well-behaved protein binders against immunotherapy targets could be very useful. Many of these receptors contain immunoglobulin (Ig) fold domains[13] with convex surfaces which can be difficult to target using binder design (Fig. 1 aand Supplementary Fig. 1).

We reasoned that a set of concave helical scaffolds with shapes tailored to interact with a wide range of convex Ig fold-containing targets could provide starting points for the design of high-stability and affinity binders to immunotherapy targets. Here, we set out to develop such scaffolds and to use them to design high-affinity binders to three key cancer immunotherapy-related receptors: TGFbRII, CTLA-4, and PD-L1. Table 1

[1]Department of Biochemistry, University of Washington, Seattle, WA, USA. [2]Institute for Protein Design, University of Washington, Seattle, WA, USA. [3]Department of Biochemistry, Albert Einstein College of Medicine, Bronx, New York, USA. [4]Department of Structural Biology, University of Pittsburgh, Pittsburgh, PA, USA. [5]Howard Hughes Medical Institute, University of Washington, Seattle, WA, USA. ✉e-mail: dabaker@uw.edu

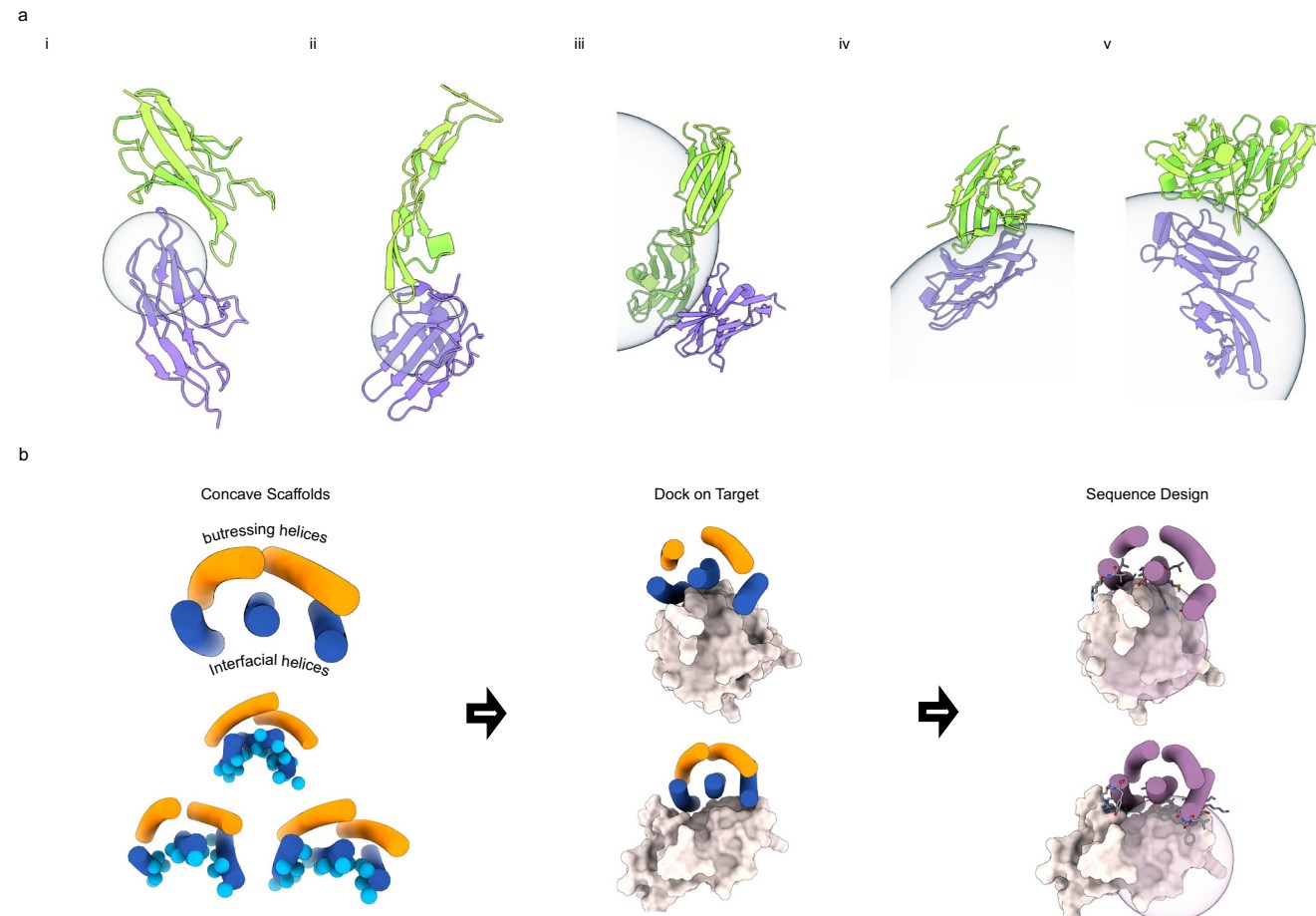

**Fig. 1 | Design of 5HCS scaffolds to target convex interfaces on immunoglobulin-like targets. a** Convex interfaces on Ig fold immune receptors. Receptors and corresponding partners are shown in purple and green cartoons, respectively. spherical surfaces fitted from interfacial heavy atoms on receptors are shown as blue transparent spheres. i, CTLA-4/CD86 complex, PDB ID: 1I85; ii, TGFβRII/TGFβ−3 complex, PDB ID: 1KTZ; iii, PD-1/PD-L1 complex, PDB ID: 3BIK;iv,

TIGIT/CD112 complex, PDB ID: 5V52; v, LAG3/LAG3-antibody complex, PDB ID: 7TZH. **b** Design workflow. Column 1: 5HCS concave scaffolds with a wide range of curvatures were designed with three helices (blue) forming the concave surfaces (Cbeta labeled as spheres) and two helices (orange) buttressing at the backside. Column 2: Docking of 5HCS scaffolds to target binding sites. Column 3: Following docking, the interface sequencing is optimized for high-affinity binding.

**Table 1 | Physicochemical properties and interface profiles of the optimized de novo 5HCS binders**

| Target | Binder ID | $K_D$ (nM) | $T_M$ (°C) | Buried surface area polar / apolar (Å$^2$) | Convexity binder / target (1/Å) |
|---|---|---|---|---|---|
| TGFβRII | 5HCS_TGFBR2_1 | < 1 | > 95 | 637.6 / 1043.2 | − 0.0669 / 0.056 |
| CTLA-4 | 5HCS_CTLA4_1 | < 0.1 | > 95 | 595.6 / 1266.1 | − 0.0593 / 0.058 |
| PD-L1 | 5HCS_PDL1_1 | 0.646 ± 0.02 | > 95 | 710.4 / 1108.9 | − 0.0310 / 0.001 |

## Results

### Computational design of 5HCS scaffolds

Naturally occurring high-affinity protein-protein interfaces exhibit significant shape complementarity, facilitating interatomic interactions and reducing solvation-free energy, which is crucial for overcoming the entropic cost of macromolecular association[14]. To systematically design binders for convex protein targets, we reasoned that scaffolds with concave shapes could be extremely useful, particularly if they have the following three properties. First, varying curvature−protein surfaces vary considerably in shape, so a set of scaffolds with diverse curvature and surface topography is ideal. Second, high stability, which provides greater tolerance to substitutions[15], allowing for more customization of the binding interface for high affinity. Third, small size−smaller scaffolds (80−120 amino acids) are more readily combinable for targeting multiple receptors, are more optimal for tumor penetration in oncology[16], and enable lower cost

gene synthesis and oligonucleotide library generation. We set out to build such a scaffold set.

Previous work with designed helical repeat proteins (DHRs) has shown that a broad range of curvatures can be achieved while maintaining high stability, but these proteins typically exceed 150 residues in length and contain 8 or more helices[17,18]. We hypothesized that reducing the number of helices could still satisfy the target properties of a concave interaction surface, tunable curvature, and high stability while reducing size. We focused on five-helix bundle scaffolds, with three helices forming the concave interface and two providing structural support (Fig. 1b). Scaffolds were generated by assembling a library of ideal helical and loop fragments into helix-turn-helix-turn modules. Each helix was constrained to 18−22 amino acids (5-6 helical turns) to balance stability and overall length. These modules were repeated three times to generate three-unit repeat proteins, and either the N- or C-terminal helix was truncated to yield five-helix proteins

under 120 amino acids. We evaluated the curvatures of the three interfacial helices and filtered out those with convex surfaces. Following Rosetta sequence design[19], we selected sequences predicted by AlphaFold2[20] to fold into the designed structures with high accuracy, as verified by DeepAccNet[21]. The selected 7476 scaffolds, referred to as 5HCS (5 helix concave scaffolds), exhibit a wide range of curvatures (Supplementary Fig. 2).

## Design and structural validation of TGFβRII binders

Protein binders to modulate TGF-β pathways have considerable interest as therapeutics in oncology, tissue fibrosis, and other areas[22]. We used the RIF-based docking protocol of Cao et al[6] to dock both the 5HCS scaffolds described above and the globular mini protein scaffold library used in the previous studies[5,23] to the binding site of transforming growth factor-β3 (TGF-β3) on the TGF-β receptor type-2 (TGFβRII)[24] (Fig. 1a). Following design and filtering[7] for binders with the concave surface of the 5HCS interacting with the target, and Alphafold2[20] based confirmation of structure and binding mode, we encoded the designs using oligonucleotide arrays and cloned into a yeast surface-expression vector to enable high throughput assessment of binding affinity. We tested in parallel 67 5HCS based designs and 4310 designs based on the globular scaffolds used by Cao et al. for comparison to our previous approach. After two rounds of fluorescent-activated cell sorting (FACS) for binding to biotinylated TGFβRII, sequencing revealed that the most enriched binders were all from the 5HCS design set (with no additional substitutions) despite the nearly 100-fold greater diversity of the minibinder library. We further optimized the most enriched design, 5HCS_TGFBR2_0, by resampling the sequences of interfacial residues in the bound state using ProteinMPNN[25] and filtering the complex models using Alphafold2[20]. We encoded the substitutions predicted to improve binding affinity in a combinatorial library using degenerate codons (Supplementary Fig. 3a) and sorted the library using yeast display selection (Supplementary Fig. 4).

Four of the most enriched optimized binders obtained after several rounds of yeast display selection were produced in *E. coli*. The highest affinity binder, 5HCS_TGFBR2_1, was found using biolayer interferometry to have an affinity less than 1 nM for TGFβRII (Fig. 2a and Supplementary Fig. 5a). The sequence identity between 5HCS_TGFBR2_0 and 5HCS_TGFBR2_1 is 88.12% (Supplementary Fig. 6a). The circular dichroism spectra indicate a helical structure with peaks at 208 nm and 222 nm consistent with the design model, and was only slightly changed by heating to 95 °C, indicating high stability (Fig. 2b).

We determined the co-crystal structures of 5HCS_TGFBR2_1 with TGFβRII. The high resolution (1.24 Å) X-ray crystal structure is very close to the computational design model (Fig. 2c; root mean square deviation (rmsd) over $C_\alpha$ atoms of 0.55 Å over the full complex), showing 5HCS_TGFBR2_1 binds to the TGF-β3 binding site on TGFβRII utilizing the concave surface as designed.

To further investigate the sequence dependence of folding and binding, we generated site saturation mutagenesis (SSM) libraries in which each residue was substituted with all other nineteen amino acids one at a time and sorted the library using FACS with fluorescent TGFβRII. Deep sequencing revealed that the interfacial and core residues, as defined by the designed model, were strongly conserved (blue indicates conservation in Fig. 2a, d, and Supplementary Fig. 7). In contrast, the surface residues were quite variable (red in Fig. 2a, d, and Supplementary Fig. 7). Helices H1, H3, and H5 which form the concave binding surface interact with TGFβRII, and the most highly conserved non-core residues are in these helices. In H1, N10 hydrogen bonds with TGFβRII D142 (Fig. 2c); in H3, S46 and S49 hydrogen bond to the backbone atoms of strand S72 - S75 (Fig. 2c); and in H5, N93 hydrogen bonds to the backbone atoms of I76 (Fig. 2d, lower panel). A hydrophobic patch composed of F48, L50, and I76 on TGFβRII critical for TGF-β3 binding packs tightly on a hydrophobic groove formed by L6 from H1, M50, V52, K53 from H3 and V96, K99, V100 from H5 (Supplementary Fig. 8). All the key interactions described above are recapitulated in the crystal structure with high side-chain orientation consistency (Fig. 2c). Design of such extended grooves and pockets is nearly impossible using small globular miniproteins; the high affinity binding and crystal structure of 5HCS_TGFBR2_1 demonstrates that 5HCS scaffolds can indeed be used to target convex binding sites.

We assessed the biological activities of 5HCS_TGFBR2_1 in cell culture signaling assays. HEK293 cells with luciferase reporter for the TGFβ SMAD2/3 signaling pathway were stimulated using 10 pM TGF-β3 and varying concentrations of 5HCS_TGFBR2_1. Dose-dependent inhibition of the TGFβ SMAD2/3 signaling was observed with an $IC_{50}$ of 30.6 nM (Fig. 2e).

## Design and structural validation of CTLA-4 binders

CTLA-4 plays an important role in peripheral tolerance and the prevention of autoimmune disease by inhibition of T cell activation. Antibody CTLA-4 targeting checkpoint inhibitors[26] have been used for melanoma and non-small cell lung cancer (NSCLC) therapy. We targeted the region surrounding the beta-turn (132–140) of CTLA-4 which is buried in the interface between CTLA-4 and CD86 (PDB ID: 1I85) (Supplementary Fig. 9a) using the methods described above. FACS of yeast libraries displaying the designs identified six CTLA-4 binders which match the designs with 100% sequence identity. Deep sequencing of a site saturation mutagenesis library of the most enriched binder, 5HCS_CTLA4_0, showed that the designed core and interfacial residues of the binder were highly conserved, suggesting the design folds and binds target as in the computational model (Fig. 3a Supplementary Fig. 9a). As the Alphafold2[20] predicted models were not consistent with the designed complex model, we combined the most enriched substitutions from the SSM heatmap, instead of using the ProteinMPNN[25] resampling followed by Alphafold2[20] filtering method described above. We again encoded these substitutions using degenerate codons (Supplementary Fig. 3b), and following yeast display selection, we expressed four of the best binders in *E. coli*. The highest affinity optimized binder, 5HCS_CTLA4_1 has a sequence identity of 82.86% compared to 5HCS_CTLA4_0 (Supplementary Fig. 6b). 5HCS_CTLA4_1 had an off rate too slow and a binding affinity for CTLA-4 too tight (<100 pM) to be measurable by biolayer interferometry (Fig. 3b and Supplementary Fig. 5b).

We determined the co-crystal structures of 5HCS_CTLA4_1 with CTLA-4 and unbound crystal structures of 5HCS_CTLA4_2 (Supplementary Fig. 10 and Supplementary Table 1). The unbound crystal structure of 5HCS_CTLA4_2 aligns with the bound structure of 5HCS_CTLA4_1 with a rmsd. of 0.416 Å. The crystal structure of 5HCS_CTLA4_1 in complex with CTLA-4 closely agrees with the design model, with a very low rmsd of 0.34 Å (Fig. 3c). 5HCS_CTLA4_1 binds to the CD86 binding site on CTLA-4 using a concave binding surface formed by H1, H3 and H5 covering both the CTLA-4 beta-turn (L98 to Y104) and hydrophobic pocket which interacts with CD86 (Supplementary Fig. 9b). H1 interacts with the hydrophobic beta-turn (L128 to Y136) through hydrophobic interactions between Y18 and M135 and aromatic interactions between H19 and Y136 (Fig. 3c, top panel). Substitution of this residue with H or Y improves binding affinity (Fig. 3c). S54 and I55 on H3 interact with Y139 on CTLA-4 (Fig. 3c, middle panel), and N89 on H5 hydrogen bonds with Q90 on CTLA-4. (Fig. 3c, lower panel). All of these interactions are closely recapitulated in the crystal structure (compare blue and green in Fig. 3c). The circular dichroism spectra indicate a helical structure with peaks at 208 nm and 222 nm consistent with the design model and was unchanged by heating to 95 °C, indicating high thermal stability (Fig. 3d).

We tested the biological activity of 5HCS_CTLA4_1 in cell culture using an immune checkpoint functional assay in which stably

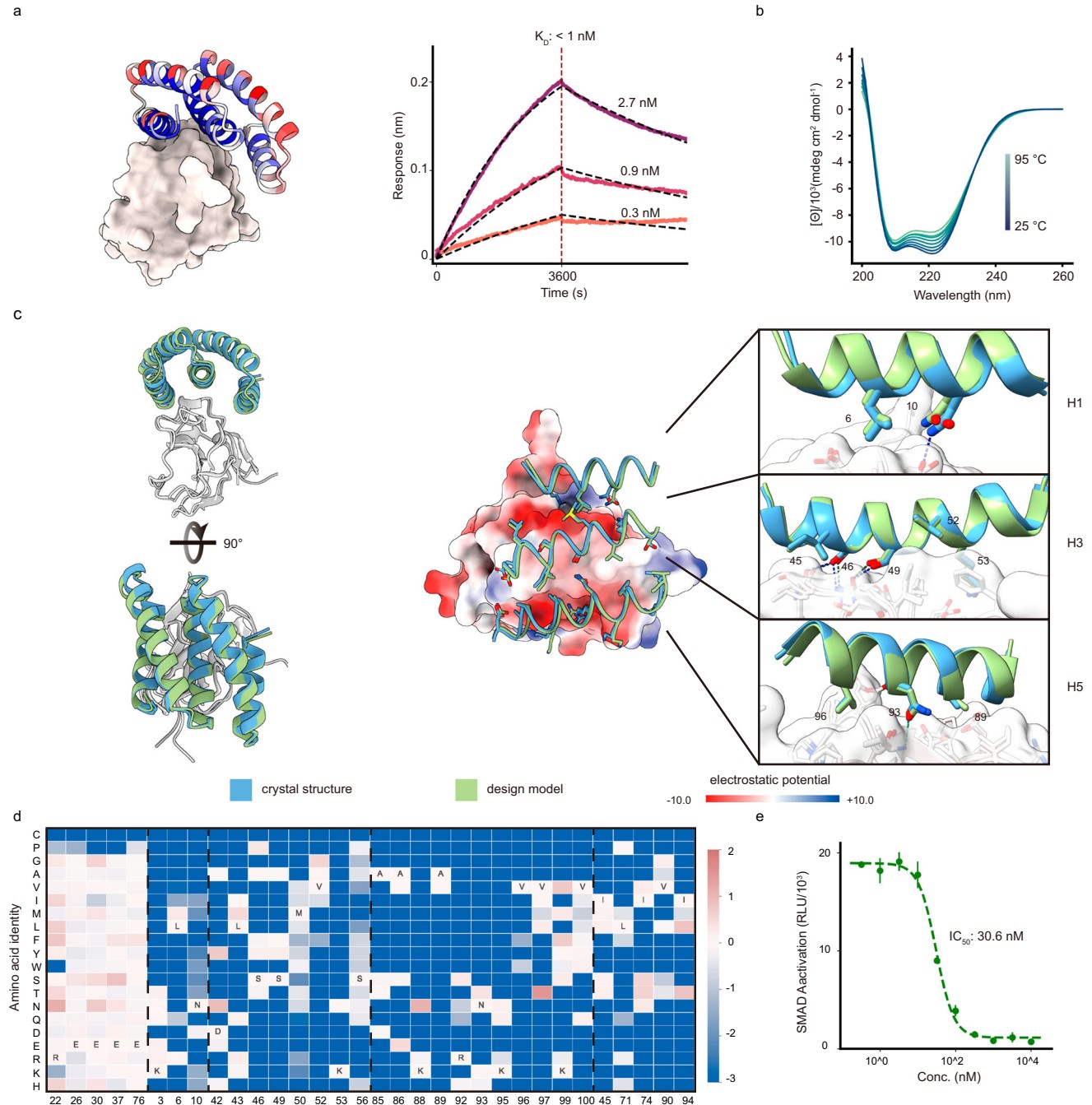

**Fig. 2 | Concave 5HCS binder to TGFβRII. a** Left: Design model of 5HCS_TGFBR2_1 (cartoon) binding to TGFβRII (PDB ID: 1KTZ). 5HCS_TGFBR2_1 is colored by Shannon entropy from the site saturation mutagenesis results at each position in blue (low entropy, conserved) to red (high entropy, not conserved). Right: Biolayer interferometry characterization of 5HCS_TGFBR2_1. Biotinylated TGFβRII were loaded to Streptavidin (SA) tips and incubated with 2.7 nM, 0.9 nM, and 0.3 nM of 5HCS_TGFBR2_1 to measure the binding affinity. The binding responses are shown in solid lines and fitted curves are shown in dotted lines. **b** Circular dichroism spectra from 25 °C to 95 °C for 5HCS_TGFBR2_1. **c** Crystal structure of 5HCS_TGFBR2_1 in complex with TGFβRII. Left are top and side views of the crystal (blue and gray) superimposed on the design models (green and white). In the middle, TGFβRII is shown in surface view and colored by electrostatic potential (using ChimeraX; red negative, blue positive). On the right, detailed interactions between 5HCS_TGFBR2_1 (blue, green) and TGFβRII (gray, white) are shown. **d** Heat map of the log enrichments for the 5HCS_TGFBR2_1 SSM library selected with 1.6 nM TGFβRII at representative positions. Enriched mutations are shown in red and depleted in blue. The annotated amino acid in each column indicates the residue from the parent sequence. **e** Dose-dependent inhibition of TGF-β3 (10 pM) signaling in HEK293 cells. The mean values were calculated from triplicates for the cell signaling inhibition assays measured in parallel, and error bars represent standard deviations. $IC_{50}$ values were fitted using four-parameter logistic regression by Python scripts. Source data (**a**, **b**, **e**) are provided in the Source Data file.

expressing CTLA-4 Jurkat cells with a luciferase reporter for TCR/CD28 activation were incubated with activating Raji cells expressing the CTLA-4 ligands CD80 and CD86. Inhibition of the inhibitory CTLA-4 - CD86 interaction results in TCR pathway activation, and hence can be directly read out using this assay. We co-cultured the cells with a range

of concentrations of the CTLA-4 binders, and observed dose-dependent activation of CTLA-4 effector cells with an $EC_{50}$ of 53.3 nM (Fig. 3e). This is higher than the $EC_{50}$ (15.0 nM) of the anti-CTLA-4 antibody Ipilimumab (MDX-010, Yervoy), despite the at least two order weaker binding affinity for CTLA-4 (18.2 nM)[27]. Steric or

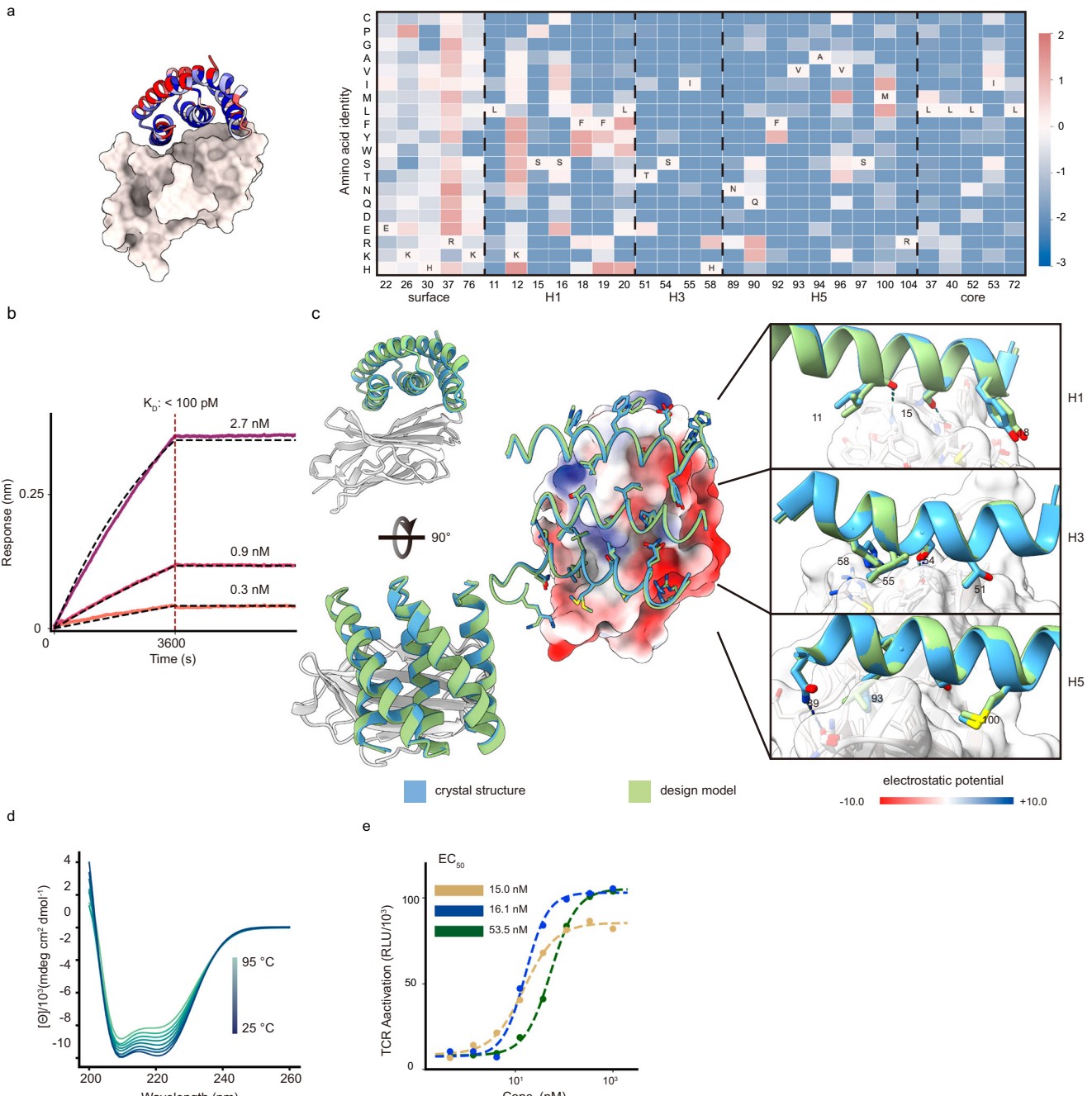

**Fig. 3 | Designed 5HCS CTLA-4 binder. a** Left: Model of 5HCS_CTLA4_1 (cartoon) binding to CTLA-4 (PDB ID:1l85) colored by Shannon entropy from site saturation mutagenesis results. Right: Log enrichments for the 5HCS_CTLA4_1 SSM library selected with 10 nM CTLA-4 at representative positions. The annotated amino acid in each column indicates the residue from the parent sequence. **b** Biolayer interferometry characterization of 5HCS_CTLA4_1. Biotinylated CTLA-4 was loaded to Streptavidin (SA) tips, and these were incubated with 2.7 nM, 0.9 nM, and 0.3 nM of 5HCS_CTLA4_1 to measure the binding affinity. **c** Crystal structure of 5HCS_CTLA4_1 in complex with CTLA-4. Color schemes are the same as Fig. 2c. Designed interactions between 5HCS_CTLA4_1 (green) and CTLA-4 (white). **d** Circular dichroism spectra from 25 °C to 95 °C for 5HCS_CTLA4_1. Color schemes and experimental details are as in Fig. 2b. **e** Increase of TCR activation induced signal (via NFAT pathway) from engineered CTLA-4 effector cells lines by 5HCS_CTLA4_1 (green), lpilimumab (gold), and 5HCS_CTLA4_1_c6 (blue) is shown. $EC_{50}$ values were fitted using four-parameter logistic regression by Python scripts. Source data (**b**, **d**, **e**) are provided in the Source Data file.

avidity effects may contribute to the potency of the antibody, which can interact with two receptors through the two Fabs. To explore the effect of avidity, we flexibly fused 5HCS_CTLA4_1 to previously designed domains which oligomerize into different symmetric architectures[28]. We found that a highly expressed and monodisperse hexameric version (Supplementary Fig. 11), 5HCS_CTLA4_1_c6 had an $EC_{50}$ of 16.1 nM, comparable to the antibody (Fig. 3e).

## Design and structural validation of PD-L1 binders

Programmed death-ligand 1 (PD-L1), is upregulated on many tumors, and interacts with PD-1 on T-cells to downregulate T-cell activation. Therapeutic antibodies against PDL1 have shown considerable promise for checkpoint inhibition in cancer immunotherapy[26]. We designed binders using the methods described above to target the binding site of PD-1 on PD-L1 (PDB ID: 3BIK) and block the interaction

between the two proteins (Fig. 1a and Supplementary Fig. 12a). Two PD-L1 binders were obtained from a set of 96 designs. We optimized the stronger binder, 5HCS_PDL1_0, by resampling the residues at the designed interface using ProteinMPNN[25] followed by Alphafold2[20] filtering. We used yeast display to sort a library with degenerate codons encoding mutations (Supplementary Fig. 3c) predicted to improve binding, expressed ten of the most enriched binders in *E. coli*, and measured their binding affinities by biolayer interferometry. The highest affinity binder, 5HCS_PDL1_1, which has 93.2% similarity with the sequence of 5HCS_PDL1_0 (Supplementary Fig. 6c), is expressed at high levels, very stable and has an affinity of 646 pM (Fig. 4a and Supplementary Fig. 5c; association traces were more regular using SPR than BLI, perhaps due to poor behavior of the recombinant native PD-L1 protein).

To examine the sequence determinants of folding and binding of 5HCS_PDL1_1 and to provide a structural footprint of the binding site, we generated an SSM library and sorted the library using FACS with fluorescent PD-L1. The conservation of both the core residues and the interfacial residues (Fig. 4a, b and Supplementary Fig. 12c) suggests the binders fold and bind to the models as designed. As with 5HCS_TGFBR2_1 and 5HCS_CTLA4_1, the interfacial helices H1, H3 and H5 of 5HCS_PDL1_1 have an overall concave shape (Fig. 4a). The key interactions between H1 and PD-L1 include aromatic packing of Y9 and Y123 on PD-L1 and electrostatic interactions between D10 and E13 with K124 and R125 on PD-L1 (Fig. 4c). H3 binds to the hydrophobic pocket formed by Y56, M115, A121, and Y123 (Fig. 4c). Residues Y9, E13, K56, and Q99 spanning the three helices satisfy the hydrogen bonding requirements of both the side chains and backbone of the PD-L1 edge beta strand (A121-R125) buried at the interface.

We solved the high-resolution crystal structure of 5HCS_PDL1_1 at 1.88 Å. The refined structure (Supplementary Table 1) reveals the expected helical assembly with five antiparallel helices (Fig. 4c and Supplementary Fig. 10b). The crystal structure of 5HCS_PDL1_1 superimposes on the computational design model with a rmsd of 0.75 Å over 105 aligned Cα atoms (Fig. 4c and Supplementary Fig. 10b; the substitutions which increase affinity relative to 5HCS_PDL1_0 do not alter the backbone structure). Not surprisingly, given the near identity between the computational designs (Supplementary Fig. 4c) and the crystal structures, the shape and electrostatic potential of the designed target binding interfaces are nearly identical between the crystal structure and the computational design model (Supplementary Fig. 10c).

To assess binder specificity on the cell surface, we stained WT and KO cell lines with high concentrations ($\geq 10 \times EC_{50}$) of the highest affinity binder, 5HCS_PDL1_1, using commercial antibodies as positive controls. FACS analysis showed minimal signal on KO cells, comparable to the negative control, indicating high target specificity of our designed binders (Fig. 4d). We tested the biological activity of 5HCS_PDL1_1 in vitro using an immune checkpoint functional assay in which stably expressing PD-1 Jurkat cells with a luciferase reporter of Nuclear factor of activated T-cells (NFAT) activity were mixed with PD-L1 expressing CHO-K1 cells to activate TCR signaling. Inhibition of the inhibitory PD-L1 - PD-1 interaction can again be read out through TCR activation using the luciferase reporter. We co-cultured the two cell lines with a range of concentrations of the PD-L1 binder and observed dose-dependent activation with an $EC_{50}$ of 2.1 nM (Fig. 4e).

## Discussion

The high affinity and potent signaling pathway modulation of the TGFβRII, PD-L1, and CTLA-4 binders described here demonstrates the considerable potential of our approach for targeting critical cell surface receptors. Thus far, small de novo-designed proteins have not exhibited high immunogenicity in humans[29], and their circulation times can be readily increased by fusion to Fc or albumin binding domains[30], which should make them useful for manipulating signaling

and checkpoint blockade. While we used Rosetta interface design methods primarily in this paper, the 5HCS scaffolds should provide excellent starting points for refinement with partial RFdiffusion[8] and other machine learning-based methods[31] which can further mold the backbone geometry to the target of interest. The high stability and modularity of the repeat protein architecture should facilitate the incorporation of conditional binding and masking logic for a next generation of immunotherapies that only function in diseased tissue to increase therapeutic benefit and reduce systemic toxicity.

## Methods

### 5HCS Scaffolds library design

**Backbone generation.** The backbones were designed by taking a library of loops and helices drawn from previous successful mini-proteins and assembling them into helix-turn-helix-turn modules of 30–50 amino acids. The modules were then repeated 3 times to give a repeat protein. All possibilities of N- and C- terminal truncation were assessed and the most concave compact structure under 120 amino acids was chosen. The backbones were diversified using the Rosetta HybrizeMover using the backbones themselves as templates.

**Sequence design and filtering.** The generated backbones were designed using standard Rosetta LayerDesign protocol[11]. The heavy atoms from the residues at the concave surfaces were selected by secondary structure and rosetta layerselector. RANSAC was used to fit spherical surfaces from the coordinates of the interfacial atoms with a threshold of 1 A and max iteration 100k. The algorithm was implemented by Python. By definition, the convexity of the surface is the reciprocal of the radius. The designed scaffolds were later filtered by the AlphaFold2 with a mean plDDT cutoff of 80 and AccNet with a mean plDDT cutoff of 0.8. There are finally 7476 scaffolds meeting all the criteria. (library availability: https://github.com/proteincraft/5HCS)

### Protein surface convexity calculation

**Protein complex structure extraction.** Pairs of interacting chains were extracted from high-quality crystals from PDB. The pairs of protein complex structures were filtered by interfacial profiles, including the length of each partner's and delta solvent accessible surface area (dSASA). Then, we clustered them 40% sequence identity on both chains and selected representatives favoring higher resolution and shorter proteins.

**Convexity calculation.** We calculated atomic SASA for both partners from protein-protein complex pairs in apo and holo structures. Heavy atoms with a difference of $0.5 A^2$ are defined as interfacial residues. RANSAC was used to fit spherical surfaces from the coordinates of the interfacial atoms with a threshold of 1 A and max iteration 100 k. The algorithm was implemented by Python. By definition, convexity of the surface is the reciprocal of the radius.

**Concave and convex definition.** To define whether the surface is concave or convex, the geometry centers of heavy atoms of the proteins and interfacial atoms were firstly calculated. The inner product of interfacial atoms centers to protein centers and interracial atoms to fitted centers was calculated. Those surfaces with minus results are defined concave vice versa. (code availability: https://github.com/proteincraft/5HCS).

### Interface design and filtering

**Docking and interface design.** For the TGFβRII binder design, the 5HCS or mini protein libraries were docked to the target binding site using patchdock[3]. Docked poses of the 5HCS library were filtered by binding orientation. Only designs with interfacial residues as the concave surfaces were kept. Interface sequence design was performed using ProteinMPNN[25] with target sequences fixed as native sequences.

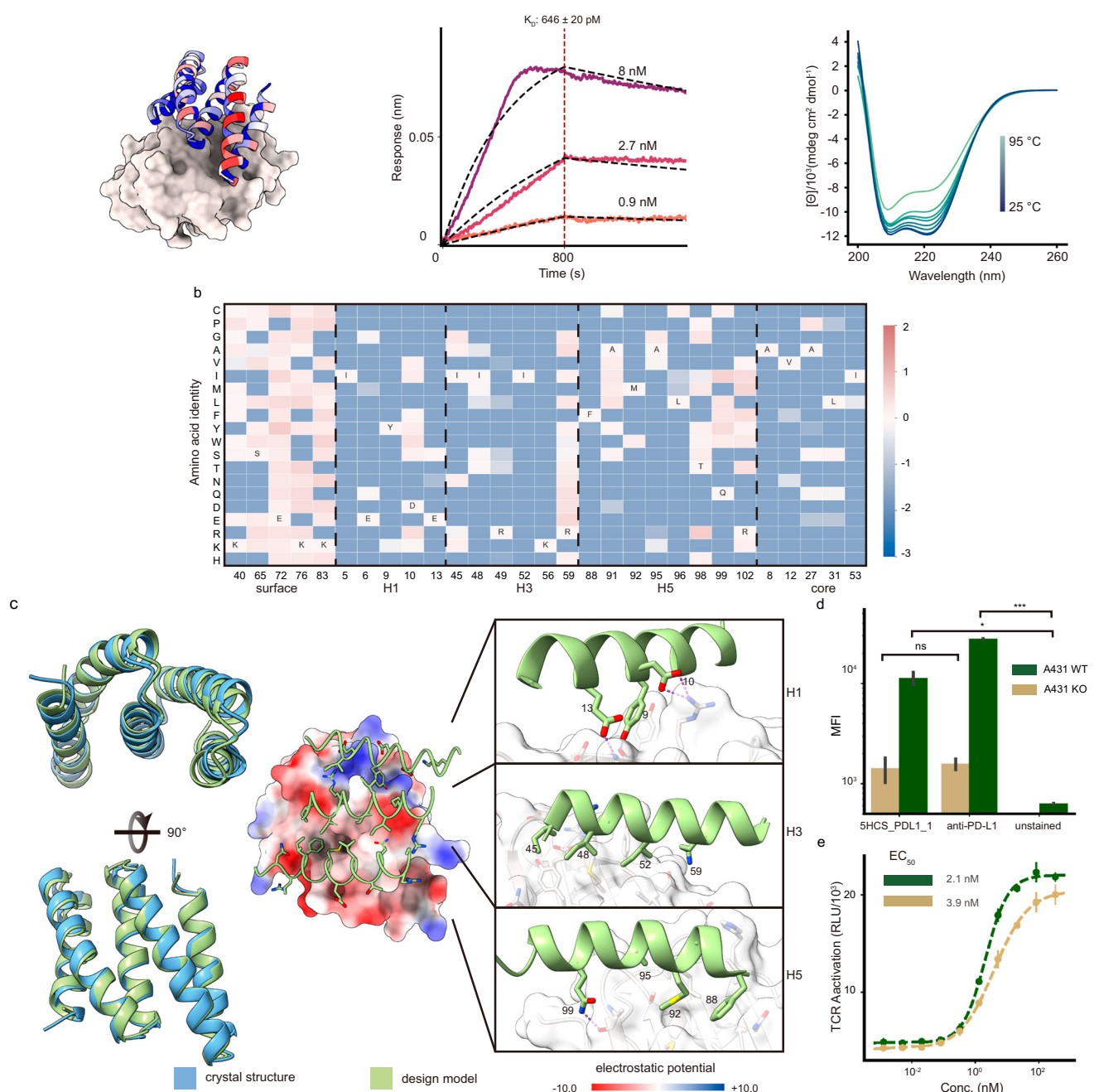

**Fig. 4 | Designed 5HCS binder to PD-L1. a** Biophysical Characterization of 5HCS_PDL1_1. Left: Model of 5HCS_PDL1_1 (cartoon) binding to PD-L1 (PDB ID: 3BIK), with 5HCS_PDL1_1 colored by Shannon entropy from site saturation mutagenesis results. Middle: Biolayer interferometry characterization of 5HCS_PDL1_1. Biotinylated PD-L1 was loaded to Streptavidin (SA) tips, and these were incubated with 8 nM, 2.7 nM, and 0.9 nM of 5HCS_PDL1_1 to measure the binding affinity. Right: Circular dichroism spectra from 25 °C to 95 °C for 5HCS_PDL1_1. **b** Heat map representing the log enrichments for the 5HCS_PDL1_1 SSM library selected with 6 nM PD-L1 at representative positions. The annotated amino acid in each column indicates the residue from the parent sequence. **c** Unbound crystal structure of 5HCS_PDL1_1 and designed interactions between 5HCS_PDL1_1 (green) and PD-L1 (white). Color schemes are the same as Fig. 2. **d** WT A431 (green) and PD-L1 KO A431 (gold) cell lines were stained with fluorophore-labeled 5HCS_PDL1_1 and anti-PD-L1

antibody respectively and then analyzed through FACS. The comparison between anti-PD-L1 vs. A431 WT and Unstain A431 WT resulted in a significance level with $p = 2.874 \times 10^{-4}$. The 5HCS_PDL1_1 vs. A431 and Unstain A431 WT comparison yielded a significant $p$-value of $p = 4.269 \times 10^{-2}$. A431 WT samples were tested in two wells, and A431 KO samples in four wells, with 5000 cells per well, ensuring reproducibility and statistical reliability. ns, no significance, *$P < 0.05$, and ***$P < 0.001$. ($t$ test independent samples with Bonferroni correction). Bars, mean ± SEM. **e** The increase of TCR activation induced signal (via NFAT pathway) from engineered PD-1 effector cell lines by 5HCS_PDL1_1 (green), control antibody (gold) is shown. The mean values were calculated from triplicates for the cell signaling inhibition assays measured in parallel, and error bars represent standard deviations. Color schemes and experimental details are as in Fig. 3. Source data (**a**, **d**, **e**) are provided in the Source Data file.

The designs were later filtered by ddG (less than -40), contact molecular surface (larger than 400), and pAE (less than 10) from AlphaFold2 initial guess[7]. Finally, 67 and 4310 designs from 5HCS and mini protein libraries passed the filters and were tested experimentally, respectively.

For the CTLA-4 binder design, the 5HCS libraries were docked to the target binding site using patchdock[6]. Docked poses of the 5HCS library were filtered by binding orientation. Only designs with interfacial residues as the concave surfaces were kept. Interface sequence design was performed using RosettaScriptsl. The designs were later filtered by ddG (less than -40), contact molecular surface (larger than 400). Finally, 4600 designs from 5HCS passed the filters and were tested experimentally.

For the PD-L1 binder design, the 5HCS libraries were docked to the target binding site using patchdock[6]. Docked poses of the 5HCS library were filtered by binding orientation. Only designs with interfacial residues as the concave surfaces were kept. Interface sequence design was performed using ProteinMPNN with target sequences fixed as native sequences. The designs were later filtered by ddG (less than -40), contact molecular surface (larger than 400), and pAE (less than 10) from AlphaFold2 initial guess[7]. Finally, 96 designs from 5HCS libraries passed the filters and were tested experimentally.

**Combinatorial library design.** The hits screened from the initial designs were further optimized by the virtual optimization protocol. Interfacial residues were re-sampled massively (5000 replicates) using ProteinMPNN[25] with a higher temperature of 0.4. As the binding pattern stays mostly the same, the re-sampled designs were later assessed by delta ddG predicted by AlphaFold2 initial guess[20]. Designs with lower ddG than the initial hits were aligned by primary sequences. At each residue position, the more times one type of mutation showed up the more likely the mutation will improve the binding affinity. We then ordered Ultramer oligonucleotides (Integrated DNA Technologies) containing the degenerate codons for the mutations predicted to be beneficial. The constructed libraries were transformed into Saccharomyces cerevisiae EBY100. The transformation efficiencies were around $10^7$.

## Yeast surface display

*Saccharomyces cerevisiae* EBY100 strain cultures were grown in C-Trp-Ura medium supplemented with 2% (w/v) glucose. For induction of expression, yeast cells were centrifuged at $4000 \times g$ for 1 min and resuspended in SGCAA medium supplemented with 0.2% (w/v) glucose at the cell density of $1 \times 10^7$ cells per ml and induced at 30 °C for 16–24 h. Cells were washed with PBSF (PBS with 1% (w/v) BSA) and labeled with biotinylated targets using two labeling methods: with-avidity and without-avidity labeling. For the with-avidity method, the cells were incubated with the biotinylated target, together with anti-c-Myc fluorescein isothiocyanate (FITC, Miltenyi Biotec) and streptavidin–phycoerythrin (SAPE, ThermoFisher). The concentration of SAPE in the with-avidity method was used at one-quarter of the concentration of the biotinylated targets (TG2-H82E4, CT4-H82E1, PD1-H82E5, acrobiosystems). For the without-avidity method, the cells were first incubated with biotinylated targets, washed, and secondarily labeled with SAPE and FITC. All the original libraries of de novo designs were sorted using the with-avidity method for the first few rounds of screening to exclude weak binder candidates, followed by several without-avidity sorts with different concentrations of targets. For SSM libraries, two rounds of without-avidity sorts were applied, and in the third round of screening, the libraries were titrated with a series of decreasing concentrations of targets to enrich mutants with beneficial mutations. The combinatorial libraries were enriched at a medium concentration of target for two rounds by collecting the top 10% of the binding population. In the third round of sorting, the enriched library was titrated to with a series of decreasing concentrations of targets.

The several binding populations with the lowest concentration of target were collected.

## Protein expression and purification

Amino acid sequences of reported in this study are provided in Supplementary Table 2. Synthetic genes were optimized for *E. coli* expression and purchased from IDT (Integrated DNA Technologies) as plasmids in the pET29b vector with a TEV-cleavable hexa-histidine affinity tag. Plasmids were transformed into BL21* (DE3) *E. coli* competent cells (Invitrogen). Single colonies from an agar plate with 100 mg/L kanamycin were inoculated in 50 mL of Studier autoinduction media 45, and the expression continued at 37 °C for over 24 h. The cells were harvested by centrifugation at $4000 \times g$ for 10 min and resuspended in a 35 mL lysis buffer of 300 mM NaCl, 25 mM Tris pH 8.0, and 1 mM PMSF. After lysis by sonication and centrifugation at $14000 \times g$ for 45 min, the supernatant was purified by $Ni^{2+}$ immobilized metal affinity chromatography (IMAC) with Ni-NTA Superflow resins (Qiagen). Resins with bound cell lysate were washed with 10 mL (bed volume 1 mL) of washing buffer (300 mM NaCl, 25 mM Tris pH 8.0, 60 mM imidazole) and eluted with 5 mL of elution buffer (300 mM NaCl, 25 mM Tris pH 8.0, 300 mM imidazole). Both soluble fractions and full cell culture were checked by SDS-PAGE. Soluble designs were further purified by size exclusion chromatography (SEC). Concentrated samples were run in 150 mM NaCl 25 mM Tris pH 8.0 on a Superdex 75 Increase 10/300 gel filtration column (Cytiva). SEC-purified designs were concentrated by 10 K concentrators (Amicon) and quantified by UV absorbance at 280 nm.

## Biolayer interferometry

Binding assays were performed on an OctetRED96 BLI system (ForteBio) using streptavidin-coated biosensors. Biosensors were equilibrated for at least 10 min in Octet buffer (10 mM Hepes pH 7.4, 150 mM NaCl, 3 mM EDTA, 0.05% Surfactant P20) supplemented with 1 mg/mL bovine serum albumin (SigmaAldrich). For each experiment, the biotinylated target protein was immobilized onto the biosensors by dipping the biosensors into a solution with 50 nM target protein for 200 to 500 s, followed by dipping in fresh octet buffer to establish a baseline for 200 s. Titrations were executed at 25 °C while rotating at 1000 rpm. The association of designs to targets on the biosensor was allowed by dipping biosensors in solutions containing designed proteins diluted in octet buffer for 800 to 3600 s. After reaching equilibrium, the biosensors were dipped into a fresh buffer solution in order to monitor the dissociation kinetics for 800–3600 s. For binding titrations, kinetic data were collected and processed using a 1:1 binding model using the data analysis software 9.1 of the manufacturer. Global kinetic fitting using three concentration data was performed for $K_D$ calculations.

## Circular dichroism

Far-ultraviolet circular dichroism measurements were carried out with a JASCO-1500 instrument equipped with a temperature-controlled multi-cell holder. Wavelength scans were measured from 260 to 190 nm at 25 and 95 °C and again at 25 °C after fast refolding (about 5 min). Temperature melts monitored the dichroism signal at 222 nm in steps of 2 °C $min^{-1}$ with 30 s of equilibration time. Wavelength scans and temperature melts were performed using 0.3 mg $ml^{-1}$ protein in PBS buffer (20 mM $NaPO_4$, 150 mM NaCl, pH 7.4) with a 1 mm path-length cuvette.

## Cell assays

**TGF-β luciferase reporter assay.** The TGF-β inhibition assays utilizing HEK-293 cells stably transfected with the $CAGA_{12}$ TGF-β reporter[32] were performed[33]. Cells were maintained in DMEM containing 10% fetal bovine serum (FBS) and 1% penicillin/streptomycin. Cells were plated at $3 \times 10^4$ cells per well in a treated 96-well plate. After 24 h, the media

was removed and replaced with fresh DMEM containing 0.1% bovine serum albumin (BSA) and a two-fold concentration series of 5HCS_TGFβR2_1. After 30 min, cells were stimulated with 10 pM TGF-β3. Twenty-four hours after stimulation, the cells were lysed, and luciferase activity was measured using luciferin. The measurements for each condition were made in triplicate. $IC_{50}$ values were calculated using the four parameters logistic regression by Python scripts.

**CTLA-4 blockade cell assay.** The CTLA-4 Blockade Bioassay (Promega) was used as described in the product literature to compare the bioactivity of our high affinity CTLA-4 binders with Ipilimumab. Briefly, 25 uL of CTLA-4 effector cells prediluted into complete RPMI media supplemented with 10% FBS were added to wells of a 96-well flat-bottomed white luminescence plate (Costar). In a separate 96-well assay plate, antibodies and binding reagents to be tested were serially diluted into RPMI media at three times the intended final concentration. The activity of the CTLA-4 binders was compared to a control hIgG antibody (Biosciences) and the FDA-approved anti-CTLA-4 mAb Ipilimumab. From this 25 μL of each diluted reagent was transferred to the wells containing CTLA-4 effector cells, and subsequently 25 μL of the aAPC/Raji Cells were also added. The resulting reactions were incubated for 16 hours at 37 °C in a humidified $CO_2$ incubator. After incubation, 75 μL of prepared Bio-Glo reagent (Promega) was added to each well, incubated for 5 min at room temperature with gentle shaking at 300 rpm and luminescence measured on an Envision plate reader (Perkin Elmer). The raw luminescence data was normalized using the following formula:

$$(RLU\ signal - background)/(RLU\ no\ antibody - background)$$

where the background and no antibody control values were each calculated from an average of three wells with no cells or cells but no antibody respectively. $EC_{50}$ values were calculated using the four parameters logistic regression by Python scripts.

**PD-L1 blockade cell assay.** The assays were performed according to the manufacturer's instructions (Promega). Briefly, PD-L1 aAPC/CHO-K1 cells were thawed in a 37 °C water bath until just thawed and transferred to pre-warmed media (90% Ham's F12 / 10% FBS). Cells were mixed and immediately seeded to the inner 60 wells of a 96-well flat bottom white cell culture plates at 100 μl volume. 100 μl of media was also added to the outside wells to prevent evaporation. Cells were incubated for 16 h in a 37 °C, 5% $CO_2$ incubator. At the end of the incubation period, 95 μl of media was removed from each of the wells. Immediately after 40 μl of appropriate antibody or binder dilutions were added to individual wells. PD-1 effector cells were thawed in similar fashion as for PD-L1 aAPC/CHO-K1 cells and transferred to pre-warmed assay buffer (99% RPMI 1640 / 1% FBS). 40 μl of PD-1 effector cells were added to the inner 60 wells of the assay plate. 80 ul of assay buffer was added to outside wells to prevent evaporation. The assay plate was incubated for 6 h in a 37 °C, 5% $CO_2$ incubator. At the end of incubation, plates were removed from the incubator and equilibrated to ambient temperature (22 - 25 °C). 80 μl of Bio-Glo reagent was added to each well and incubated for 10 mins. Luminescence was measured using the BioTek Synergy Neo2 multi-mode reader. $EC_{50}$ values were calculated using the four parameters logistic regression by Python scripts.

**Specificity determination**
**Cell surface receptor knockouts.** A431 cells had PD-L1 knocked out via CRISPR RNP transfection. RNP complexes were formed by incubating 4 μl of 80 μM guide RNA (IDT guides: Hs.Cas9.CD274.1.AA, Hs.Cas9.CD274.1.AB) with 4 μl of 80 uM tracrRNA (IDT cat. 1072533) at 37 °C for 30 min. To generate complete RNPs, 4 μl of 40 μM guide complex was incubated with 4 μl of 40 μM cas9-NLS (Berkeley

MacroLab) at 37 °C for 30 min. For electroporation, $2 \times 10^5$ cells of each cell type in 20 μl of electroporation buffer (Lonza, cell line SF for A431) were mixed with 1 μl of electroporation enhancer (IDT cat. 1075916) and 2 ul of assembled RNPs prior to loading 20 μl into an electroporation cuvette strip (Lonza cat. V4XC-2032). Cells were electroporated with appropriate settings (A431:EQ-100). Cells were immediately rescued with warm complete media and transferred to a 24-well plate to grow after resting for 5 min at 37 °C with 5% $CO_2$. Cells were tested for knockout efficiency by TIDE analysis. Genomic DNA was extracted with Lucigen Quickextract (Lucigen cat. QE0905T) and amplified with NEBNext high-fidelity polymerase (NEB cat. M0541S).

**Cellular surface staining.** A431 cells were stained with PD-L1 binder or antibody to compare the specificity of de novo binders to commercial antibodies. For staining, $5 \times 10^5$ cells were washed twice with 200 μl cell staining buffer (Biolegend cat. 420201) in a 96 well u-bottom plate. Cells were then resuspended in 50 μl of staining mixture (cell staining buffer and fluorophore-conjugated binder or antibody (Biolegend cat. 329713) and incubated on ice in the dark for 30 min. Cells were washed three times with 200 μl staining buffer and then analyzed on a ThermoFisher Attune.

**Structure determination**
**Expression and purification.** The coding sequence for residues 46–155 of human TβRII (UniProt P37173) was inserted into plasmid pET32a (EMD-Millipore) between the NdeI and HindIII sites without inclusion of any expression tags, transformed into chemically competent *E. coli* BL21(DE3) (EMD-Millipore), expressed at 37 °C in the form of insoluble inclusion bodies, and refolded and purified to homogeneity[13]. The 5HCS_TGFBR2_1 used for crystallization was prepared as described above, followed by digestion for 12 h at 25 °C with TEV protease (1:25 mass ratio) in 25 mM Tris, 100 mM Tris, pH 8.0, 1 mM DTT, 1 mM EDTA. The identity of the isolated protein products was verified by measuring their intact masses, which were found to be within 0.5 Da of the calculated masses (Thermo UltiMate UHPLC coupled to Bruker Compact QqTOF ESI quadrupole TOF mass spectrometer). The TbRII:5HCS_TGFBR2_1 complex was isolated by size exclusion chromatography using a HiLoad Superdex 75 26/60 column (GE Healthcare, Piscataway, NJ) in 25 mM HEPES pH 7.5, 100 mM NaCl at a 1:1.1 ratio, with 5HCS_TGFBR2_1 being in slight excess. The complex peak fractions were pooled and concentrated to 33 mg/mL for crystal screening.

For large-scale purifications of the CTLA-4 and PD-L1 binders for crystallization, 2-liter bacterial cultures were grown in Super Broth (Teknova) media supplemented with antibiotics and antifoam 204 (Sigma) at 37 °C in LEX 48 airlift bioreactors (Epiphyte3, Canada) to an A600 of 3. The temperature was then reduced to 22 °C, isopropyl-β-D-thiogalactoside (IPTG) was added to 0.5 mM, and the cultures were incubated overnight. Cells were harvested by centrifugation at $14,000 \times g$ and suspended in buffer containing 20 mM HEPES (pH 7.5), 500 mM NaCl, 20 mM imidazole, 0.1% IGEPAL, 20% sucrose, 1 mM β-mercaptoethanol (BME). Cells were disrupted by sonication, and debris was removed by centrifugation at $45,000 \times g$. The supernatants were applied to a chromatography column packed with 10 ml His60 SuperFlow resin (Clontech Laboratories) that had been equilibrated with buffer A (50 mM HEPES pH 7.5, 30 mM imidazole, 500 mM NaCl, and 1 mM BME). The columns were washed with buffer A, and the $His_6$-binder proteins were eluted with buffer B (20 mM HEPES, pH 7.5, 350 mM NaCl, 400 mM imidazole, and 1 mM BME). The $His_6$ tags were removed by overnight digestion at 4 °C with the TEV protease at a 1500:1 ratio of binder:TEV. The tag-free binders were then separated from $His_6$-tags by Superdex 200 gel filtration equilibrated with a buffer containing 20 mM HEPES pH 7.5, 350 mM NaCl. The CTLA-4 and PD-L1 binders migrated through gel filtration as discrete peaks with estimated molecular weights of 14 kDa and 12 kDa, respectively, indicating

that they are monomers in solution. The purity of the binders was judged by SDS-PAGE and Coomassie blue staining. The peak fractions from the gel filtration step were pooled and concentrated to 20–25 mg/ml in a buffer containing 20 mM HEPES (pH 7.5) and 150 mM NaCl. 5HCS_CTLA4_1cb:CTLA-4 complex were purified using size exclusion chromatography (Superdex S200) equilibrated with a buffer containing 20 mM HEPES pH 7.5, 150 mM NaCl. The peak fractions were pooled and concentrated to 7.5 mg/ml The preparations were flash frozen in liquid nitrogen and stored at − 80 °C for long-term storage.

### Protein crystallization and crystal harvesting
Crystals of the TβRII:5HCS_TGFBR2_1 complex were formed using hanging drop vapor diffusion in 24-well plates with 300 μL of well solution and siliconized glass cover slips. Crystals formed in 1–2 days at 25 °C with drops prepared by mixing 0.4 μL 25 mg/mL protein complex and 0.4 μL of 20% (w/v) PEG-MME 5 K, 0.4 M $(NH_4)_2$ $SO_4$, 0.1 M Tris pH 7.4, and 16 – 32 % glycerol. The crystals were mounted in nylon loops without additional cryoprotectants and with excess well solution wicked off.

Screening of 5HCS_CTLA4_1cb2 and 5HCS_PDL1_1 for crystal formation was performed using 0.8 μL (protein: reservoir solution = 1:1) crystallization drops at a concentration of 15 mg/ml with a Crystal Gryphon (Art Robbins Instruments) robot, using MCSG (Microlytic), Index HT, Crystal Screen HT, and Peg Ion HT sparse matrix crystallization suites (Hampton Research). Initial crystals obtained from the sparse matrix screening were further optimized with several rounds of grid screening using a Formulator robot (Art Robbins Instruments). Optimized crystallization conditions for diffraction quality binder protein crystals and cryo-protectants used during crystal harvesting are summarized in Supplementary Table 1.

### Data collection and processing, structure refinement and analysis
The diffraction data for the TβRII:5HCS_TGFBR2_1 complex was collected at the Southeast Regional Collaborative Access Team (SER-CAT) 22-ID beamline at the Advanced Photon Source, Argonne National Laboratory. The data was integrated with XDS[14] and the space group ($P2_12_12_1$ with dimensions a,b,c = 47.98 Å, 57.17 Å, 78.80 Å and α,β,γ = 90°, 90°, 90°) was confirmed via pointless[34]. The data was reduced with aimless[35], ctruncate[36–40] and the uniquify script in the CCP4 software suite[41]. Phasing was performed with Phaser[42], initially with the 1.1 Å TβRII X-ray structure (PDB 1M9Z), followed by the predicted 5HCS_TGFBR2_1 structure. Several cycles of refinement using Refmac5[43–50] and model building using COOT[51] were performed to determine the final structure (Supplementary Fig. 13). Data collection and refinement statistics are shown in Supplementary Table 1.

Data from the crystals of CTLA-4 binder were collected on a Dectris Pilatus 6 M detector, with a wavelength of 0.98 Å, on the ID-31 (LRL-CAT) beamline at the Argonne National Laboratory (Supplementary Table 1). Single crystal data were integrated and scaled using iMosflm[52] and aimless[41], respectively. Diffraction was consistent with the orthorhombic space group $P2_12_12_1$ (unit cell dimensions are in Supplementary Table 1) and extended to 1.85 Å resolution with one molecule (chain A) in the asymmetric unit. Data for the PD-L1 binder crystals were collected on a Dectris EIGER X 9 M detector, with a wavelength of 0.92 Å, on the 17-ID-1 (AMX) beamline at the Brookhaven National Laboratory (Supplementary Table 1). Data for the CTLA-4-binder complex crystals were collected on a Dectris EIGER X 9 M detector, with a wavelength of 0.98 Å, on the 17-ID-2 (FMX) beamline at the Brookhaven National Laboratory (Supplementary Table 1). The datasets were indexed, integrated, and scaled using fastDP, XDS[53] and aimless[52], respectively. The PD-L1 crystals belong to the tetragonal space group and diffracted to 1.88 Å. The CTLA-4-binder complex crystal belongs to C2 space group and diffracted to 2.72 Å. Initial

phases of 5HCS_CTLA4_1cb2, 5HCS_PDL1_1, and 5HCS_CTLA4_1cb:CTLA-4 complex were determined by molecular replacement (MR) with Phaser[42]. using coordinates of the computationally designed respective binders and binder complex; the initial MR coordinate was manually inspected and corrected using Coot[51]. The model was refined with Phenix-Refine[54]. Analyses of the structures were performed in Coot and evaluated using MolProbity[55]; B-factors were calculated using the Baverage program in CCP4 suite[41]. The crystallographic model exhibited excellent geometry with no residues in disallowed regions of the Ramachandran plot[56] (Supplementary Fig. 14). Crystallographic statistics and RCSB accession codes are provided in Supplementary Table 1. All figures depicting structure were generated with PyMol unless stated otherwise.

### Reporting summary
Further information on research design is available in the Nature Portfolio Reporting Summary linked to this article.

## Data availability
The X-ray crystallography coordinates and structure files data generated in this study have been deposited in the Protein Data Bank under the following accession codes: 8G4K (5HCS_TGBR2_1 complex), 8GAB (5HCS_CTLA4_1 complex), 8GAC (5HCS_CTLA4_2), 8GAD (5HCS_PDL1_1). Miseq data from combinatorial libraries are deposited at PRJNA975208. Source data are provided in this paper.

## Code availability
The code used to carry out protein design in this study is also available at https://doi.org/10.5281/zenodo.14775883[57].

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

## Acknowledgements

Funding for this work was provided by a gift from Gates Ventures (W.Y., L.S., D.B.), the National Institute on Aging (NIA) Grants R01AG063845 (I.G., D.B.) and U19AG065156 (D.R.H.), the Audacious Project at the Institute for Protein Design (A.A., S.F.H., D.L., C.J.K., S.R.G., D.B.), grants from DARPA supporting the Harnessing Enzymatic Activity for Lifesaving Remedies (HEALR) program (HR001120S0052 contract HR0011-21-2-0012, A.K.B., B.H., D.B.) and the Synergistic Discovery and Design project (HR001117S0003 contract FA8750-17-C-0219, L.C., D.B.), the Open Philanthropy Project Improving Protein Design Fund (Y.W., D.B.), and the Howard Hughes Medical Institute (B.C., D.B.).

## Author contributions

W.Y. and D.B. designed the research. W.Y., B.C., and D.H. designed the 5HCS scaffold library. W.Y., D.H., B. H., and L.C. designed the binders. W.Y., D.H., I.G., S.H., and A.A. performed library preparation, yeast screening, expression, and binding experiments. A.G. solved the structure of the bound and unbound CTLA-4 binder. A.G., A.H. solved the structure of the unbound PD-L1 binder. T.A.S and A.H solved the structure of the bound TGFbRII binder. W.Y., D.H., Z.L., S.G., P. H. and A. M. expressed and purified proteins. W.Y. and Y.W. performed circular dichroism measurements. C.H., D.L. performed a TGFb3 inhibition assay. S.G. and C.K. performed a CTLA-4 activation assay. C.K. performed a PD-L1 activation assay. W.Y., D.H., and S.R. performed binding assays. T. S. and N. E. design oligomer scaffolds. All authors analyzed data. L.S. and D.B. supervised research. W.Y. and D.B. wrote the manuscript. All authors revised the manuscript.

## Competing interests

David Baker, Wei Yang, Derrick Hicks, Brian Coventry, and Inna Goreshnik have filed a patent application related to the minibinders described in this work. The authors declare no other competing interests.
