## [Transparent Peer Review file · Nature Communications]

Design of High-Affinity Binders to Immune Modulating Receptors for Cancer Immunotherapy

Corresponding Author: Professor David Baker

Version 0:

Reviewer comments:

Reviewer #1

(Remarks to the Author)

In this manuscript, Yang and colleagues present their work on designed helical concave scaffold proteins targeting three immune receptors. This work is a focused example of de novo protein design, a topic well explored in their previous publications. The study spans computational design, biological assays, and validation through co-crystal structure determination.

From a drug design perspective, readers of Nat Comm will appreciate the scope of the molecular targets—three receptors—which demonstrates the generality of the approach for identifying ligands for convex target sites.

Overall, this is interesting and relevant work. I support its publication in Nat Comm after minor revisions.

Minor comments:

Figure 4c: The BLI sensogram for PD-L1 binders shows a low signal (~0.08 nm), and the on-rate part of the curve appears unusual with a disproportionate binding response and fitted curve. The authors should comment on this.

Methods: target protein biotinylation method should be described (referenced)

(Remarks on code availability)

Reviewer #2

(Remarks to the Author)

Baker et al use here pre-existing modules that could form concave structures to assemble possible "mini protein" binders to known receptors (with well understood structures). After protein design based on these pre-existing structures, to select the best binders by presenting them within the context of a yeast surface display library using soluble receptor as bait. The top binders are then made recombinantly within E. coli and validation ensues (using standard methods).

This is a solid paper with a great idea. The experimental work appears very solid from the bioinformatics, experimental structural validation, to cellular assays. The main question I have is how generalizable this method is? I think addressing this (and the immunogenicity issue mentioned below) would flesh out the discussion.

- what happens if structures are not known? would an Alphafold representation be accurate enough as a start to lead to the high affinities needed?
- what happens to the system if key miniproteins are lost because they are not amenable to yeast display? (what are the blind spots)?
- what happens to the system if the targets cannot be purified to put on the octet biosensors for miniprotein binder selection?
- is this all predicated on a priori knowledge of how these target proteins interact with their receptors? and what if there are no receptors? (eg MUC-1)?

Finally, what of the long-term issue for even successfully designed binders from the perspective of host immune response? Would artificial proteins as therapeutics potentially lead to problematic immune reactions, both for the individual and the treatment (i.e., clearance of the protein)?

(Remarks on code availability)

Reviewer #3

(Remarks to the Author)

Yang and colleagues present work on computational design of binders to TGFBR2, PD-L1, and CTLA4. 5-helix proteins and globular miniproteins were evaluated with the helical designs performing more effectively. Lead molecules were affinity matured and characterized for binding, structure, and biological activity (in one elementary assay for each). The manuscript adds to the growing list of computationally designed binders. The targets have therapeutic relevance. Thus, the work could be of interest to numerous readers. Yet, the utility / impact of the manuscript is moderate. There are several opportunities to strengthen insight: [1] provide more detailed analysis on the comparison of 5HCS vs. globular miniproteins; [2] evaluate the biological mechanism and/or efficacy of the engineered binders more thoroughly; and/or [3] gain insight about the binder design approach to strengthen future design efforts. In addition, there are numerous statements that need stronger support either via analysis of current data or citation of relevant literature. Details follow:

- The enrichment of 5HCS binders relative to globular miniprotein binders is an intriguing result, but the details are not clear. Were the number of variants of each design comparable? Were the depth of designs comparable? Further evaluation of this comparison would greatly aid the breadth of utility of the manuscript.
- The statement, “The current designs provide new routes for manipulating signaling and checkpoint blockade” is not supported by the data. The molecules exhibit manipulation but not new mechanisms. In general, the biological evaluation of the lead molecules is minimal. It'd be valuable to explore mechanism and/or capabilities more thoroughly.
- It would be advantageous to evaluate if any insight can be gained by comparing the designs that were effective versus those that were ineffective. How can the current study aid future efforts?
- The assertion, “These novel protein binders exhibit superior properties versus conventional modalities with high affinity, high thermal stability, and more versatile functions.9–12” is not clearly supported by the cited papers. The affinities are not higher than many conventional binders, and “superior” stability and versatility are not clearly evident in the papers. To be clear, affinity, stability, and versatility are present, but the assertion of superiority is not clear.
- Support the assertion, “Deep sequencing results were closely consistent with the design model” with data. For example, show the correlation between prediction and experimental results.
- Numerous assertions should be supported by referenced literature (or data):
 1. “convex surfaces which can be more difficult to target using binder design”
 2. “the more stable the scaffold, the more customization can occur at the binding interface, leading to robust, high-affinity binders.”
 3. “smaller scaffolds (80-120 amino acids) are ... advantageous for applications like tumor penetration in oncology.”

(Remarks on code availability)

Version 1:

Reviewer comments:

Reviewer #1

(Remarks to the Author)

Authors have improved the manuscript according to the suggestions of the reviewers. I have no additional comments and I support publication in Nat Comm.

(Remarks on code availability)

Reviewer #2

(Remarks to the Author)

My concerns and questions have been addressed.

(Remarks on code availability)

Reviewer #3

(Remarks to the Author)

The revised manuscript has addressed the critiques.

(Remarks on code availability)

Point-to-Point Response to Reviewer Comments:

We would like to thank the editor for guidance and coordinating the review of this manuscript, and the Reviewers for their thoughtful comments. We have carefully addressed the issues raised by the Reviewers in a point-to-point manner, as presented below in non-italicized blue text. We also include for convenience the relevant revised content of the manuscript in italicized blue text beneath each response. We have also made very minor text edits throughout to improve clarity (not highlighted).

Reviewer1:

In this manuscript, Yang and colleagues present their work on designed helical concave scaffold proteins targeting three immune receptors. This work is a focused example of de novo protein design, a topic well explored in their previous publications. The study spans computational design, biological assays, and validation through co-crystal structure determination.

From a drug design perspective, readers of Nat Comm will appreciate the scope of the molecular targets—three receptors—which demonstrates the generality of the approach for identifying ligands for convex target sites.

Overall, this is interesting and relevant work. I support its publication in Nat Comm after minor revisions.

We would like to thank the reviewer for the appreciative comments on the quality and impact of our work and constructive suggestions.

Minor comments:

Figure 4c: The BLI sensogram for PD-L1 binders shows a low signal (~ 0.08 nm), and the on-rate part of the curve appears unusual with a disproportionate binding response and fitted curve. The authors should comment on this.

Methods: target protein biotinylation method should be described (referenced)

Response: We agree with this criticism and added an explanation about the traces that could not be well fit on line 219. We add the source of the biotinylated target proteins on line 427.

Line 219:

association traces were more regular using SPR than BLI, perhaps due to poor behavior of the recombinant native PD-L1 protein

Line 427:

The concentration of SAPE in the with-avidity method was used at one-quarter of the concentration of the biotinylated targets (TG2-H82E4, CT4-H82E1, PD1-H82E5, Acrobiosystems).

Reviewer #2 (Remarks to the Author):

Baker et al use here pre-existing modules that could form concave structures to assemble possible "mini protein" binders to known receptors (with well understood structures). After protein design based on these pre-existing structures, to select the best binders by presenting them within the context of a yeast surface display library using soluble receptor as bait. The top binders are then made recombinantly within E. coli and validation ensues ! (using standard methods).

This is a solid paper with a great idea. The experimental work appears very solid from the bioinformatics, experimental structural validation, to cellular assays.

We would like to thank the reviewer for the appreciative comments on the quality and impact of our work. We also agree that it would strengthen the paper if we were to add a paragraph that addresses the questions that the Reviewer has asked. We provide our detailed response below each question and following the last question we provide in italicized text the paragraph that we have added to the Discussion.

The main question I have is how generalizable this method is? I think addressing this (and the immunogenicity issue mentioned below) would flesh out the discussion.

- what happens if structures are not known? would an Alphafold representation be accurate enough as a start to lead to the high affinities needed?

Response: From our experience on other targets, a highly accurate predicted target structure from Alphafold2 or 3 can be used to computationally design binders with high-affinity to the selected target.

- what happens to the system if key miniproteins are lost because they are not amenable to yeast display? (what are the blind spots)?

Response: Loss of functional miniproteins is possible as the reviewer suggests, for example, due to limitations in oligonucleotide library synthesis accuracy and the potential inability of some miniproteins to be effectively displayed on yeast. We believe this is not negatively impacting binder selection with the procedures that we use since we have performed next-generation sequencing (NGS) to evaluate the coverage of the designed miniproteins. Typically, our libraries achieve up to 99% coverage (after sorting for expression), ensuring a broad representation of our designs. Despite these measures, the success rate of computationally designed binders remains around 5%, which is consistent with typical benchmarks¹. Even in rare instances where a key miniprotein might be lost, our strategy ensures that functionally equivalent designs are likely to be identified.

- what happens to the system if the targets cannot be purified to put on the octet biosensors for miniprotein binder selection?

Response: The system that we use requires the purified target with commonly used tags for labeling, such as biotin, His-tag, or Fc-tag. However, if the targets themselves cannot be purified, one solution could be resurfacing the target first but keeping the functional site unchanged. An example of this strategy is designing soluble GPCRs which can be used for ligand screening². Another solution people sometimes use is to display the receptors on yeast or mammalian cells and use these receptor-expressing cells for screening³.

- is this all predicated on a priori knowledge of how these target proteins interact with their receptors? and what if there are no receptors? (eg MUC-1)?

Response: Binders were designed using the information from target proteins alone. To design a binder against a given binding site, prior knowledge of native ligands is not needed. However, whether the binder can function as an antagonist or agonist is determined by the binding site being picked which has to come from prior knowledge. Taking human MUC-1 as an example, there is only an alphafold-predicted structure available for the folded domains, the high pLDDT region from 1050 to 1150 can be used as a target to design binders against.

Finally, what of the long-term ! issue for even successfully designed binders from the perspective of host immune response? Would artificial proteins as therapeutics potentially lead to problematic immune reactions, both for the individual and the treatment (i.e., clearance of the protein)?

Response: There are not many results about the immune response of de novo-designed proteins. In the case of neo2, which was under clinical study, low immune responses were reported^{4,5}. The reason may be our de novo miniproteins are highly stable, typically with melting temperatures approaching 100 °C, which reduces the chances for APC uptake and stimulation of an immune response.

Small proteins are cleared from circulation rather fast through glomerular filtration in the kidneys. However, the half-life of circulation for small proteins can be extended by several methods including lipidation, binding to albumin, FC fusion, or PEGylation⁶.

We added this information to the discussion to clarify the potential of these binders as therapeutics.

Line 250:

Thus far, small de novo designed proteins have not exhibited high immunogenicity in humans, and their circulation times can be readily increased by fusion to Fc or albumin binding domains, which should make them useful for manipulating signaling and checkpoint blockade.

Reviewer #3 (Remarks to the Author):

Yang and colleagues present work on computational design of binders to TGFBR2, PD-L1, and CTLA4. 5-helix proteins and globular miniproteins were evaluated with the helical designs performing more effectively. Lead molecules were affinity matured and characterized for binding, structure, and biological activity (in one elementary assay for

each). The manuscript adds to the growing list of computationally designed binders. The targets have therapeutic relevance. Thus, the work could be of interest to numerous readers. Yet, the utility / impact of the manuscript is moderate. There are several opportunities to strengthen insight: [1] provide more detailed analysis on the comparison of 5HCS vs. globular miniproteins; [2] evaluate the biological mechanism and/or efficacy of the engineered binders more thoroughly; and/or [3] gain insight about the binder design approach to strengthen future design efforts. In addition, there are numerous statements that need stronger support either via analysis of current data or citation of relevant literature. Details follow:

We would like to thank the reviewer for the appreciative comments on the quality and impact of our work and constructive suggestions.

- The enrichment of 5HCS binders relative to globular miniprotein binders is an intriguing result, but the details are not clear. Were the number of variants of each design comparable? Were the depth of designs comparable? Further evaluation of this comparison would greatly aid the breadth of utility of the manuscript.

Response: We have rephrased the text to clarify how the experiments were conducted. The globular minibinders and 5HCS were two separate libraries, and both were sorted against the same target protein. The docking and design methods were identical, ensuring that the number of variants for each scaffold and the depth of each design were comparable. However, the globular library contained at least 10 times more scaffolds than the 5HCS library. Despite this, the success rate of the 5HCS library was 2 out of 67, whereas the globular library yielded no hits out of 4,310. This difference highlights the significantly higher success rate of the 5HCS library, even though more designs were sampled and tested in the globular library.

Line 102:

We tested in parallel 67 5HCS based designs, and 4,310 designs based on the globular scaffolds used by Cao et al for comparison to our previous approach. After two rounds of fluorescent-activated cell sorting (FACS) for binding to biotinylated TGFβRII, sequencing revealed that the most enriched binders were all from the 5HCS design set (with no additional substitutions) despite the nearly 100-fold greater diversity of the minibinder library.

- The statement, “The current designs provide new routes for manipulating signaling and checkpoint blockade” is not supported by the data. The molecules exhibit manipulation but not new mechanisms. In general, the biological evaluation of the lead molecules is minimal. It’d be valuable to explore mechanism and/or capabilities more thoroughly.

Response: We apologize for the potentially misleading use of the term “new routes” since in this study no new mechanisms of inhibition were developed. We have revised the sentence to clarify this point. We sincerely appreciate the reviewer’s valuable suggestion to investigate in greater detail the advantages and disadvantages of the inhibitors we have developed compared to existing inhibitors, although this is beyond the scope of the current paper.

Line 250:

Thus far, small de novo designed proteins have not exhibited high immunogenicity in humans, and their circulation times can be readily increased by fusion to Fc or albumin binding domains, which should make them useful for manipulating signaling and checkpoint blockade.

- It would be advantageous to evaluate if any insight can be gained by comparing the designs that were effective versus those that were ineffective. How can the current study aid future efforts?

Response:

We thank the Reviewer for their suggestion and agree that gaining insights from this study would be highly beneficial. We attempted to analyze the models tested in this study using the cases of TGF β RII and PD-L1, as CTLA-4 binder complexes could not be predicted by AlphaFold2, which is currently regarded as one of the most important metrics. As shown below, for most of the metrics used to evaluate the designs, the values of hits (solid lines) fall within the range of average \pm standard deviation for at least one of the targets. For some metrics, the hits even trend in the opposite direction. This suggests that we did not sample a sufficient number of binders in this study to allow for meaningful statistical analysis. Typically, when aiming to identify principles of binder design, we generate over 10,000 designs for a single target. However, in this study, we only screened about 100 designs, focusing instead on identifying binders rather than exploring fundamental principles. While scaling up might have increased the chances of illuminating the key principles of successful binder design, the small number of binders tested in this study limits the feasibility of conducting a robust and conclusive analysis.

Solid lines represent the hits identified for FACS sorting for corresponding targets. The dotted lines indicate the range of the mean \pm standard deviation (average \pm std) of the distribution shown. Scores displayed in green indicate that higher values are better, while scores in purple indicate that lower values are preferable. Among the scores plotted here: 1) pae_binder, pae_interaction, and plddt_binder, which are AlphaFold scores indicating the confidence of folding and binding; 2) contact_molecular_surface, hydrophobic_residue_contacts, interface_sc and interface_buried_sasa, which indicate the contacts between binder and target; 3) ddg, ddg_hydrophobic, hbond_bb_sc, hbond_sc are scores, which indicate the interaction free energy; 4) postsap_score, presap_score, score_per_res, ss_sc and total_score, which are general Rosetta scores indicating the solubility and energetics of the proteins.

- The assertion, “These novel protein binders exhibit superior properties versus conventional modalities with high affinity, high thermal stability, and more versatile functions.9–12” is not clearly supported by the cited papers. The affinities are not higher than many conventional binders, and “superior” stability and versatility are not clearly evident in the papers. To be clear, affinity, stability, and versatility are present, but the assertion of superiority is not clear.

Response: We agree with the Reviewer and have modified our sentence to eliminate the comparison to existing inhibitors for these target receptors.

Line 47:

The development of computational protein design has enabled the creation of de novo protein binders against human cell surface receptors with high modularity, affinity, and stability.

- Support the assertion, “Deep sequencing results were closely consistent with the design model” with data. For example, show the correlation between prediction and experimental results.

Response: We thank the Reviewer for pointing out this vague sentence. To clarify, the deep sequencing results here referred to the enrichment ratio of all possible single-point mutations of the binder. The results show that interfacial or core residues of the designed binders are more conserved (more blue in Fig. S6.) compared to surface residues (more red in Fig. S6). We corrected the expression below.

Line 130:

Deep sequencing revealed that the interfacial and core residues, as defined by the designed model, were strongly conserved (blue indicates conservation in Fig. 2 a,e, and S6). In contrast, the surface residues were quite variable (red in Fig. 2 a,e, and S6).

- Numerous assertions should be supported by referenced literature (or data):
 1. “convex surfaces which can be more difficult to target using binder design”
 2. “the more stable the scaffold, the more customization can occur at the binding interface, leading to robust, high-affinity binders.”
 3. “smaller scaffolds (80-120 amino acids) are ... advantageous for applications like tumor penetration in oncology.”

Response: We agree with the Reviewer that these assertions should be better supported by references or data.

1. Regarding Point #1, We added a plot to Fig S1 to support the claim. In Fig S1, we showed that previously designed mini binders are flat or convex, and bind to flat or concave targets (Fig. S1a). We further showed by systematically analyzing the convexities of protein-protein interfaces from the PDB that the convexity of the two binding partners for each complex is negatively correlated: when one partner is convex, the other is almost always concave (Fig. S1a). We, therefore, assert that it will be more difficult to engage convex targets with current binders, as these are flat or convex and thus lack concave shapes that are needed, as with natural proteins, to effectively engage.

2. Regarding point #2, the core idea behind this is that the greater the stability of the scaffold, the greater the tolerance to mutations to introduce functional sites. This concept was previously demonstrated in studies utilizing the Sac7d protein, in which the authors found a greater number of mutations could be introduced to enable functionality as the stability of the scaffold was increased⁷. We have rephrased the sentence to better articulate this important point.

Line 68:

Second, high stability, which provides greater tolerance to substitutions, allowing for more customization of the binding interface for high affinity.

3. Antibodies and other large proteins have limited tissue and tumor penetration due to their large size, while smaller proteins have demonstrated enhanced tumor penetration⁸. We have added the reference (ref 16) to the main text as well.

References

1. Cao, L. *et al.* Design of protein-binding proteins from the target structure alone. *Nature* **605**, 551–560 (2022).
2. Goverde, C. A. *et al.* Computational design of soluble and functional membrane protein analogues. *Nature* **631**, 449–458 (2024).
3. Krohl, P. J. *et al.* Discovery of antibodies targeting multipass transmembrane proteins using a suspension cell-based evolutionary approach. *Cell Rep. Methods* **3**, 100429 (2023).
4. Silva, D.-A. *et al.* De novo design of potent and selective mimics of IL-2 and IL-15. *Nature* **565**, 186 (2019).
5. Case, J. B. *et al.* Ultrapotent miniproteins targeting the SARS-CoV-2 receptor-binding domain protect against infection and disease. *Cell Host Microbe* **29**, 1151-1161.e5 (2021).
6. Zaman, R. *et al.* Current strategies in extending half-lives of therapeutic proteins. *J. Controlled Release* **301**, 176–189 (2019).
7. Béhar, G. *et al.* Tolerance of the archaeal Sac7d scaffold protein to alternative library designs: characterization of anti-immunoglobulin G Affitins. *Protein Eng. Des. Sel.* **26**, 267–275 (2013).
8. Vazquez-Lombardi, R. *et al.* Challenges and opportunities for non-antibody scaffold drugs. *Drug Discov. Today* **20**, 1271–1283 (2015).